# Can Machine Learning Models Detect and Predict Lymph Node Involvement in Prostate Cancer? A Comprehensive Systematic Review

**DOI:** 10.3390/jcm12227032

**Published:** 2023-11-10

**Authors:** Eliodoro Faiella, Federica Vaccarino, Raffaele Ragone, Giulia D’Amone, Vincenzo Cirimele, Claudia Lucia Piccolo, Daniele Vertulli, Rosario Francesco Grasso, Bruno Beomonte Zobel, Domiziana Santucci

**Affiliations:** Radiology Department, Fondazione Policlinico Universitario Campus Bio-Medico, 00128 Roma, Italy; e.faiella@policlinicocampus.it (E.F.); federica.vaccarino@unicampus.it (F.V.); raffaele.ragone@unicampus.it (R.R.); giulia.damone@unicampus.it (G.D.); v.cirimele@policlinicocampus.it (V.C.); c.piccolo@policlinicocampus.it (C.L.P.); daniele.vertulli@unicampus.it (D.V.); r.grasso@policlinicocampus.it (R.F.G.); b.zobel@policlinicocampus.it (B.B.Z.)

**Keywords:** radiomics, lymph nodes, prostate cancer, artificial intelligence, machine learning

## Abstract

(1) Background: Recently, Artificial Intelligence (AI)-based models have been investigated for lymph node involvement (LNI) detection and prediction in Prostate cancer (PCa) patients, in order to reduce surgical risks and improve patient outcomes. This review aims to gather and analyze the few studies available in the literature to examine their initial findings. (2) Methods: Two reviewers conducted independently a search of MEDLINE databases, identifying articles exploring AI’s role in PCa LNI. Sixteen studies were selected, and their methodological quality was appraised using the Radiomics Quality Score. (3) Results: AI models in Magnetic Resonance Imaging (MRI)-based studies exhibited comparable LNI prediction accuracy to standard nomograms. Computed Tomography (CT)-based and Positron Emission Tomography (PET)-CT models demonstrated high diagnostic and prognostic results. (4) Conclusions: AI models showed promising results in LN metastasis prediction and detection in PCa patients. Limitations of the reviewed studies encompass retrospective design, non-standardization, manual segmentation, and limited studies and participants. Further research is crucial to enhance AI tools’ effectiveness in this area.

## 1. Introduction

Prostate cancer (PCa) is a significant global health concern, with a 13.5% incidence rate and a 6.7% mortality rate, making it the fifth leading cause of death in men worldwide [1]. The prognostic outlook for patients with PCa is primarily influenced by the presence or absence of distant metastases, with lymph node metastases being a particularly important prognostic factor. Various researchers have examined the distinct morphological features of lymph node metastases to identify the most significant prognostic characteristics. Certain features, such as the involvement of multiple lymph nodes compared to a single node or the volume of cancer within the lymph nodes, have demonstrated a robust prognostic value [2]. The recurrence of LNI (lymph node involvement) in patients initially diagnosed with PCa is reported to be between 1.3% and 12% and is strongly associated with mortality [3]. Currently, the definition of nodal status is determined through Prostate-specific membrane antigen Positron Emission Tomography/Computed Tomography (PSMA PET/CT), Magnetic Resonance Imaging (MRI), or Computed Tomography (CT). For both MRI and CT, the size is the only criterion adopted for adenopathy, with a short-axis diameter cut-off of 1 cm [4].

The European Association of Urology (EAU) and National Comprehensive Cancer Network (NCCN) guidelines have recommended the use of various nomograms, including the Briganti [5] and Memorial Sloan Kettering Cancer Center (MSKCC) nomograms, to aid clinicians in identifying PCa patients who may benefit from extended lymph node dissection (ePLND) during radical prostatectomy (RP) and those patients who may avoid lymphadenectomy.

For intermediate- and high-risk prostate cancer patients, the EAU recommends ePLND or pelvic lymph node dissection (PLND) if the estimated risk of positive lymph nodes (LNs) exceeds 5% [6]. Currently, PLND is the most accurate staging procedure to detect pelvic lymph node metastasis in these patients [7]. However, performing PLND during RP is associated with a higher risk of complications, including the development of lymphocele (the incidence of symptomatic lymphocele is approximately 8%) [8], bleeding, infection, and damage to nearby structures, such as nerves and blood vessels.

In order to minimize the surgical risk in these patients and improve patient outcome, in recent years non-invasive methods, which may determine the LN status with a certain accuracy, have been investigated. Artificial Intelligence (AI), particularly Machine Learning (ML), algorithms have been deeply employed in diagnostic and prognostic steps of the oncological field as a decision support tool.

Both radiomics and Deep Learning (DL) algorithms allow the extraction and analysis of quantitative imaging features from radiographic images. In particular, Radiomics algorithms require human intervention for feature extraction, while DL algorithms are human-independent and exploit mathematical computational processes to find the most accurate answer for a specific outcome. The resulting information can be used to develop descriptive and predictive models, combining image features and phenotypes with anamnestic data or gene and protein signatures (Radiogenomics) [9,10,11]. Recently, AI algorithms have emerged as a promising tool in the diagnosis and detection of various tumors [12,13], including clinically significant prostate cancer, for staging and evaluating treatment response [14]. However, despite the existence of some studies in the literature regarding lymph node involvement detected by AI tools [15,16], only a few studies have investigated the predictive ability of radiomics for lymph node invasion in PCa, and no specific reviews on this topic have been conducted.

This review aims to gather all the papers published to date concerning the role of AI in lymph node involvement detection and prediction in PCa patients, analyzing various imaging methodologies and examining their preliminary findings.

## 2. Methodology

Two radiologists independently selected the studies and extracted data from each study using MEDLINE databases, such as PubMed and Web of Science, and the following strings: (prostate) AND (radiomics OR Machine OR Deep OR Artificial) AND (lymph).

No limitations were applied to the search strategy. Inclusion Criteria: Only studies that included lymph node involvement in patients with PCa and utilized at least one imaging technique among CT, MRI, and PET-CT were included. Additionally, only publications in English, French, or Spanish with full-text availability and studies involving adult participants (aged 18 and older) were considered for inclusion.

Exclusion Criteria: Abstracts, reviews, case studies, letters to editors, comments, and editorials were excluded. Furthermore, studies involving pediatric populations were excluded. The last update of our database search was completed in August 2023.

From a total of 192 papers, 16 research articles were considered eligible; of these, three studies were prospective, and the others were all retrospective.

Eight studies were MRI-based, two were CT-based, and six were PET-CT-based (PSMA, [18F]DCFPyL, and 18F-FMCH PET-CT). The radiomics quality score (RQS) was used to evaluate the quality of the methodology.

For each study, the following information was collected: title, authors, publication year and journal, study design (retrospective or prospective), number of patients, imaging technique (TC, MRI, PET-CT), software utilized for segmentation and feature selection, algorithms employed for classification, and resulting accuracy.

In the process of composing the review, we adhered to the PRISMA statement guidelines; the checklist is available for reference in the Appendix A and the flow diagram for the selection of the studies included is summarized in Figure 1.

## 3. Results

Our search found 16 publications on lymph node metastasis prediction or detection in PCa patients using AI systems considered eligible for inclusion in the review. All these studies were published from 2019 to 2023. The study characteristics are shown in Table 1. Details about Machine Learning algorithms employed by each paper are reported in Table 2.

### 3.1. mpMRI

In their recent paper, Liu X. et al. developed and tested radiomics models based on ADC maps for pelvic LNI prediction in PCa. The radiomics model proposed that utilized an automatically segmented prostate mask achieved a higher area under the curve (AUC), with a value of 0.73, compared to the Briganti 2017, the MSKCC nomograms, and the PI-RADS assessment models, which showed AUCs of 0.69, 0.71, and 0.70, respectively. These results showed comparable accuracy of the radiomics model proposed to the current MKSCC and Briganti 2017 [17].

Furthermore, the same research group explored the use of diffusion-weighted imaging (DWI)-based radiomics and demonstrated that a prediction model of LNI combining radiological and radiomics features achieved a higher AUC (0.83, 95% CI: 0.76–0.89) compared to a model that utilized only quantitative radiological LNI features [18].

Zheng H. et al. proposed a Multiparametric MRI-Based Imaging radiomics model to predict the likelihood of pelvic LNI in PCa patients, which showed an impressive AUC of 0.915 (95% CI: 0.846–0.984) in the testing set, surpassing existing nomograms with AUCs ranging from 0.698 to 0.724 (*p* < 0.05) [19].

The study conducted by Bourbonne V. et al. in 2021, aimed to develop and validate an LNI risk prediction model in PCa using radiomics features extracted from preoperative multimodal MRI; radiomics features were extracted from the index tumor volumes, and a prediction model was trained using a neural network approach combining clinical, radiomics, and all other imaging features. The proposed model resulted in a C-Index of 0.89 in the testing set, which was higher than currently available models such as Partin, Roach, Yale, MSKCC, and Briganti 2012 and 2017 (C-Index of 0.71, 0.66, 0.55, 0.67, and 0.65 and 0.73, respectively) [20].

Liu X. et al. also investigated the feasibility of using the 3D U-Net algorithm for automated detection and segmentation of lymph nodes on pelvic DWI images in patients with suspected PCa. The accuracy of the algorithm was evaluated using various metrics, including the Dice score, the positive predictive value (PPV), the true positive rate (TPR), the volumetric similarity (VS), the Hausdorff distance (HD), the Average distance (AVD), and the Mahalanobis distance (MHD) against manual annotations of pelvic LNs. The AUC of the model to detect suspicious lymph nodes in PCa patients was 0.963 (95% CI: 0.892–0.993), indicating high accuracy [21].

In a recent study, Xiang Liu et al. developed and evaluated a DL algorithm for the semi-automated assessment of treatment response of pelvic LNs in patients with advanced PCa [22]. The algorithm used a previously reported DL model to perform automated segmentation of the LN, and its performance was evaluated using the Dice similarity coefficient (DSC) and volumetric similarity (VS). The automated segmentation-based assessment demonstrated high accuracies of 0.92 (95% CI: 0.85–0.96), 0.91 (95% CI: 0.86–0.95), and 75% (95% CI: 0.46–0.92) for target lesions, nontarget lesions, and nonpathological lesions, respectively, comparable to that of the radiologist’s scores. The DL-based algorithm has the potential to assist in the evaluation of treatment response according to MET-RADS-P criteria, which is an important but time-consuming task for patients with advanced PCa.

In 2019 Hou Y. et al. developed an ML-assisted model to identify PCa patients who require ePLND by integrating clinical, biopsy, and MRI findings. ML-based models demonstrated higher predictive accuracy than the MSKCC nomogram and improved risk prediction [23].

In another study, Hou Y. et al. developed a PLNM-risk calculator to accurately identify pelvic LN metastasis in PCa patients, integrating radiologists’ interpretations, clinicopathological factors, and refined imaging indicators from MR images using radiomics ML and deep transfer learning algorithms. The PLNM-Risk model showed strong diagnostic accuracy with AUCs of 0.93, 0.92, and 0.76 in the training/validation, internal test, and external test groups, respectively, and spared a greater number of ePLNDs (59.6% compared to MSKCC’s 44.9% and Briganti’s 38.9%), while also producing fewer false positives (59.3% compared to MSKCC’s 70.1% and Briganti’s 72.7%) [24].

### 3.2. CT and CT/PET

Franzese et al., in a retrospective study conducted in 2022, assessed the performance of radiomics models derived from non-contrast CT treatment planning series in predicting metastatic progression among high-risk PCa patients, by integrating radiomics features with clinical characteristics. The researchers utilized a predictive model to classify patients into subgroups with favorable or unfavorable prognoses, with a threshold selected using the Youden method. The study investigated the performance of pure clinical models, pure radiomics models, and a combined predictive model. The combined predictive model, incorporating both clinical and radiomics features, demonstrated superior predictive performance compared to the pure clinical and pure radiomics models [25].

In 2020, Peeken JC. et al. developed a radiomics model using CT to predict the status of lymph node metastases (LNM) in patients with recurrent PCa. The study included 80 patients who underwent prostate-specific membrane antigen (PSMA) radioguided surgery for the removal of PSMA PET/CT-positive LNM. The radiomics model combined texture, shape, intensity, and local binary pattern features and achieved a significantly better predictive performance than conventional CT parameters, with an AUC of 0.95. The model outperformed conventional CT parameters, with LN short diameter, LN volume, and expert rating achieving AUCs of 0.84, 0.80, and 0.67, respectively [26].

In their prospective study, Zamboglou C. et al. evaluated the performance of radiomics features derived from PSMA PET for non-invasive PCa discrimination and characterization of its biological properties. Their results showed that radiomics features derived from PSMA PET were effective in distinguishing between cancerous and non-cancerous tissue in the prostate. Furthermore, the texture feature “QSZHGE” was able to effectively differentiate between Gleason Score 7 and ≥8 tumors (AUC = 0.91 in the prospective cohort and AUC = 0.84 in the validation cohort) as well as between patients with nodal spread (pN1) and non-nodal spread (pN0) disease (AUC = 0.87 in the prospective cohort and AUC = 0.85 in the validation cohort) [27]. Cysouw MCF. et al. prospectively evaluated 76 intermediate- to high-risk PCa patients scheduled for prostatectomy using [18F]DCFPyL PET-CT. Radiomics features were extracted from primary tumors using random forest models to predict the presence of metastasis, LNI, Gleason score ≥ 8, and extracapsular extension (ECE). Compared to standard PET metrics, the ML models based on radiomics successfully predicted LNI (AUC 0.86 ± 0.15, *p* < 0.01), nodal or distant metastasis (AUC 0.86 ± 0.14, *p* < 0.01), Gleason score (0.81 ± 0.16, *p* < 0.01), and ECE (0.76 ± 0.12, *p* < 0.01); radiomics-based models had higher AUCs for predicting LNI and metastasis [28].

In a recent paper, Trägårdh E. et al. created and validated an automated AI-based method for detecting and quantifying PCa, LNI, and bone metastases using [18F]PSMA-1007 PET-CT images from 660 patients. The performance of the AI method was evaluated by comparing it with manual segmentations conducted by multiple nuclear medicine physicians. The AI method demonstrated an average sensitivity of 79% in detecting prostate tumor/recurrence, 79% for LNM, and 62% for bone metastases, while the corresponding average sensitivities for nuclear medicine physicians were 78%, 78%, and 59%, respectively [29]. The same research group developed and validated an AI-based method for detecting pelvic LNM in high-risk PCa using [18F]PSMA PET-CT scans in 211 patients in another similar study, with an additional goal of making the method accessible to other researchers. The AI method achieved an 82% sensitivity in identifying pelvic LNM (with an average of 1.8 false positives per patient), while the average sensitivity of human readers was 77% [30]. Moreover, Borrelli P. et al. developed and validated an AI-based tool for the detection of LN lesions in patients with PCa using 18F-choline PET/CT scans, and explored its association with PCa-specific survival. The AI-based tool demonstrated similar performance to that of experienced readers in detecting LNI (98 vs. 87/117 for Reader B, *p* = 0.045; 90 vs. 87/111 for Reader A, *p* = 0.63), and the number of detected lesions, curative treatment, and PSA were significantly linked to PCa-specific survival [31]. Furthermore, in a prospective study by Sollini et al., patients with recurrent PCa after primary radical therapy who were eligible for [18F]FMCH PET/CT scans were invited to participate in a prospective observational trial. In this trial, radiomic analysis was employed to assess intra-patient lesion similarity in both oligometastatic and plurimetastatic PCa cases. The study revealed comparable lesion heterogeneity between patients with up to five lesions and plurimetastatic patients. Patients with up to three lesions exhibited lower heterogeneity compared to plurimetastatic patients [32].

**Table 1 jcm-12-07032-t001:** Study characteristics.

Title	Authors	Publication Year	Journal	Type	Number of Patients	Imaging Techniques	AIM	SEGM	AI	RQS	Best Results	Conclusions
Preoperative prediction of pelvic lymph nodes metastasis in prostate cancer using an ADC-based radiomics model: comparison with clinical nomograms and PI-RADS assessment.	Liu X. et al. [17]	2022	Abdom Radiol (NY)	Retrospective	474 (between 2017 and 2020). Another cohort of 128 Between 2020 and 2021.	3T mpMRI	develop and test radiomics models based on manually corrected or automatically gained masks on ADC maps for PLNM prediction in Pca patients	2d–3d U-Net automatic	AUC and DCA for comparison between the optimal radiomics model and MSKCC, Briganti 2017 nomograms, and PI-RADS assessment	16	The radiomics model based on the mask of automatically segmented prostate obtained the highest AUC among the four radiomics models (0.73 vs. 0.63 vs. 0.70 vs. 0.56). Briganti 2017, MSKCC nomograms, and PI-RADS assessment-yielded AUCs of 0.69, 0.71, and 0.70, respectively, and no significant differences were found compared with the optimal radiomics model (*p* = 0.605–0.955).	The radiomics model based on the mask of automatically segmented prostate offers a non-invasive method to predict PLNM for patients with PCa. It shows comparable accuracy to the current MKSCC and Briganti nomograms.
Utility of diffusion weighted imaging-based radiomics nomogram to predict pelvic lymph nodes metastasis in prostate cancer.	Liu X. et al. [18]	2022	BMC Med Imaging	Retrospective	84	3T mpMRI	DWI-based radiomics for preoperative PLNM prediction in PCa patients	3d U-Net of LN in DWI	Two preoperative PLNM prediction models with quantitative radiological LN features alone (Model 1) and combining radiological and radiomics features (Model 2) via multiple logistic regression. The visual assessments of junior (Model 3) and senior (Model 4) radiologists were compared. C-index of the nomogram analysis and DCA were used to evaluate performance and clinical usefulness.	17	No significant difference between AUCs of Models 1 and 2 (0.89 vs. 0.90; *p* = 0.573) in the held-out cohort. Model 2 showed the highest AUC (0.83, 95% CI 0.76, 0.89) for PLNM prediction in the LN subgroup with a short diameter ≤ 10 mm compared with Model 1 (0.78, 95% CI 0.70, 0.84), Model 3 (0.66, 95% CI 0.52, 0.77), and Model 4 (0.74, 95% CI 0.66, 0.88). The nomograms of Models 1 and 2 yielded C-index values of 0.804 and 0.910, respectively, in the held-out cohort. The C-index of the nomogram analysis (0.91) and decision curve analysis (DCA) curves confirmed the clinical usefulness and benefit of Model 2.	A DWI-based radiomics nomogram incorporating the LN radiomics signature with quantitative radiological features is promising for PLNM prediction in PCa patients, particularly for normal-sized LNM
Multiparametric MRI-based radiomics model to predict pelvic lymph node invasion for patients with prostate cancer	Zheng H. et al. [19]	2022	Eur Radiol.	Retrospective	244	3T mpMRI	predicting lymph node invasion (LNI), via a radiomics-based machine learning approach.	manual segm of index lesion on T2w seq and ADC map	An integrative radiomics model (IRM) (SVM) performance was measured by AUC, sensitivity, specificity, NPV, and PPV.	16	Overall, 17 (10.6%) and 14 (16.7%) patients with LNI were included in training/validation set and testing set, respectively. Shape and first-order radiomics features showed usefulness in building the IRM (integrative radiomics model). The proposed IRM achieved an AUC of 0.915 (95% CI: 0.846–0.984) in the testing set, superior to pre-existing nomograms whose AUCs were from 0.698 to 0.724 (*p* < 0.05).	The proposed IRM could be potentially feasible to predict the risk of having LNI for patients with PCa. With the improved predictability, it could be utilized to assess which patients with PCa could safely avoid ePLND, thus reducing the number of unnecessary ePLND
Development of a Radiomic-Based Model Predicting Lymph Node Involvement in Prostate Cancer Patients	Bourbonne V. et al. [20]	2021	Cancers (Basel)	Retrospective	280	3T e 1.5 T MRI	develop and internally validate a novel LNI risk prediction model based on radiomic features extracted from preoperative multimodal MRI	The index lesion was a semi-automatic 3D Slicer.	Radiomic features were extracted from the tumor volumes in ADC map and T2 seq (ComBat harmonization method for inter-site heterogeneity). A prediction model was trained using a neural network approach (Multilayer Perceptron Network) combining clinical, radiomic and all features. AUC and the C-Index for performance	17	The proposed model resulted in a C-Index of 0.89 in the testing set, which was higher than currently available models such as Partin, Roach, Yale, MSKCC, and Briganti 2012 and 2017.	The study suggests that radiomic features extracted from preoperative MRI scans combined with clinical features through a neural network could improve LNI risk prediction in PCa.
Radiomics-based prognosis classification for high-risk prostate cancer treated with radiotherapy	Franzese C. et al. [25]	2022	Strahlenther Onkol	Retrospective	157	Non-contrast CT	risk of metastatic progression in HR Pca patients after RT and ADT, considering MFS	Manual of gland only, prostate gland + seminal vesicles, seminal vesicles only.	5 clinical features + 62 radiomics features were combined using R platform	13		Radiomic features were able to predict the risk of metastatic progression in high-risk prostate cancer. Combining the radiomic features and clinical characteristics can classify high-risk patients into favorable and unfavorable prognostic groups
A CT-based radiomics model to detect prostate cancer lymph node metastases in PSMA radioguided surgery patients	Peeken JC. et al. [26]	2020	Eur J Nucl Med Mol Imaging	Retrospective	80	Contrast-CT	CT-based radiomic model to predict LNM status using a PSMA radioguided surgery (RGS) cohort with histological confirmation of all suspected lymph nodes (LNs).	LN segmentation was conducted manually using Eclipse 13.0 on the contrast-enhanced diagnostic CT.	156 radiomic features analyzing texture, shape, intensity, and LBP were extracted. Least absolute shrinkage and selection operator (radiomic models) and logistic regression (conventional parameters) were used for modeling	19	A combined radiomic model achieved the best predictive performance with a testing AUC of 0.95	The best radiomic model outperformed conventional measures for detection of LNM demonstrating an incremental value of radiomic features.
[18F]FMCH PET/CT biomarkers and similarity analysis to refine the definition of oligometastatic prostate cancer	Sollini M. et al. [32]	2021	EJNMMI Res.	Prospective	92	[18F]FMCH PET/CT	evaluate [18F]FMCH PET/CT radiomic analysis in patients with recurrent PCa after primary radical therapy, testing intra-patient lesions similarity in oligometastatic and plurimetastatic PCa, comparing the two most used definitions of oligometastatic disease	Lesions were semi-automatically segmented by the PET VCAR software (GE Healthcare, Waukesha, WI, USA) on a General Electric workstation	supervised and supervised clustering of patients for identifying the prevalence of metastasis	15		They found a comparable heterogeneity between patients with up to five lesions and plurimetastatic patients, while patients with up to three lesions were less heterogeneous than plurimetastatic patients, featuring different cell phenotypes in the two groups. Our results supported the use of a 3-lesion threshold to define oligometastatic PCa.
Radiomic features from PSMA PET for non-invasive intraprostatic tumor discrimination and characterization in patients with intermediate- and high-risk prostate cancer–a comparison study with histology reference	Zamboglou C. et al. [27]	2019	Theranostic	Prospective	20	PSMA PET	tumor discrimination and non-invasive characterization of Gleason score (GS) and pelvic lymph node status.	manually created segmentations of the intraprostatic tumor volume	Coregistered histopathological gross tumor volume. 133 radiomics feat from GTV-Histo of Pca and non-Pca. AUC evaluated the discrimination of Gleason and LN involvement	23	The texture feature QSZHGE was a statistically significant (*p* < 0.01) predictor for PCa patients with GS ≥ 8 tumors and pN1 status.	Radiomic features derived from PSMA PET discriminated between PCa and non-PCa tissue within the prostate. Additionally, the texture feature QSZHGE discriminated between GS 7 and GS ≥8 tumors and between patients with pN1 and pN0 disease.
Machine learning-based analysis of [18F]DCFPyL PET radiomics for risk stratification in primary prostate cancer	Cysouw MCF et al. [28]	2021	Eur J Nucl Med Mol Imaging	Prospective	76	[18F]DCFPyL PET-CT	ability of ML-based analysis of quantitative [18F]DCFPyL PET metrics to predict metastatic disease or high-risk pathological tumor features	primary tumors were delineated using 50–70% peak isocontour thresholds on images with and without partial-volume correction	408 extracted feat analyzed with RF; Gleason, LNI, mets, ECE. Models were trained with PET feat alone and radiomics ones	24	The radiomics-based machine learning models predicted LNI (AUC 0.86 ± 0.15, *p* < 0.01), nodal or distant metastasis (AUC 0.86 ± 0.14, *p* < 0.01), Gleason score (0.81 ± 0.16, *p* < 0.01), and ECE (0.76 ± 0.12, *p* < 0.01). The highest AUCs reached using standard PET metrics were lower than those of radiomics-based models.	Machine-learning-based analysis of quantitative [18F]DCFPyL PET metrics can predict LNI and high-risk pathological tumor features in primary PCa patients. These findings indicate that PSMA expression detected on PET is related to both primary tumor histopathology and metastatic tendency.
A machine learning-assisted decision-support model to better identify patients with prostate cancer requiring an extended pelvic lymph node dissection	Hou Y. et al. [23]	2019	BJU Int.	Retrospective	248	3T mpMRI	develop an ML-assisted model for identifying the candidates for ePLND in PCa by integrating clinical, biopsy, and precisely defined MRI findings	No radiomic features	18 integrated features using LR, SVM, and RF models were compared to an MSKCC nomogram. Performance evaluated with AUC and DCA	7	The predictive accuracy of ML-based models, with or without MRI-reported LNI, yielded similar AUCs, which were higher than the MSKCC nomogram (0.816; *p*-value < 0.001).	ML-based model below 15% cutoff, superior to MSKCC nomogram, allows up to 50% or more ePLNDs spared at the cost of missing < 3% LNIs.
Integration of clinicopathologic identification and deep transferrable image feature representation improves predictions of lymph node metastasis in prostate cancer	Hou Y et al. [24]	2021	EBioMedicine	Retrospective	401	3TmpMRI	PLNM-Risk calculator to obtain a precisely informed decision about whether to perform extended pelvic lymph node dissection	PCa lesions manually segmented	set of radiologists’ interpretations, clinicopathological factors and newly refined imaging indicators from MR images with radiomics machine learning and deep transfer learning algorithms. Its clinical applicability was compared with Briganti and Memorial Sloan Kettering Cancer Center (MSKCC) nomograms.	14	PLNM-Risk achieved good diagnostic discrimination with areas under the receiver operating characteristic curve (AUCs) of 0.93 (95% CI, 0.90–0.96), 0.92 (95% CI, 0.84–0.97) and 0.76 (95% CI, 0.62–0.87) in the training/validation, internal test and external test cohorts, respectively.	PLNM-Risk calculator offers a noninvasive clinical biomarker to predict PLNM for patients with PCa.
Freely Available, Fully Automated AI-Based Analysis of Primary Tumour and Metastases of Prostate Cancer in Whole-Body [18F]-PSMA-1007 PET-CT.	Trägårdh E et al. [29]	2022	Diagnostics (Basel)	Retrospective	660	[18F]PSMA PET-CT	develop and validate a fully automated AI- based method for detection and quantification of suspected prostate tumor/local recurrence, LN, and bone mets from [18F]PSMA-1007 PET-CT	manual	evaluation of total lesion volume and uptake by CNN vs. nuclear medicine physician	7	sensitivity of the AI method: 79% for detecting prostate tumor/recurrence, 79% for lymph node metastases, and 62% for bone metastases. On average, nuclear medicine physicians’ corresponding sensitivities were 78%, 78%, and 59%, respectively.	The development of an AI-based method for prostate cancer detection with sensitivity on par with nuclear medicine physicians was possible.
Semiautomated pelvic lymph node treatment response evaluation for patients with advanced prostate cancer: based on MET-RADS-P guidelines	Liu X et al. [20]	2023	Cancer Imaging.	Retrospective	162	3T mpMRI	Deep-learning-based algorithm for semiautomated treatment response assessment of pelvic lymph nodes.	Automated segmentation of pelvic lymph nodes.	The performance of the deep learning algorithm was evaluated using the Dice similarity coefficient (DSC) and volumetric similarity (VS). Kappa statistics were used to assess the accuracy and consistency of the treatment response assessment by the deep learning model and two radiologists	8	The accuracies of automated segmentation-based assessment were 0.92 (95% CI: 0.85–0.96), 0.91 (95% CI: 0.86–0.95), and 75% (95% CI: 0.46–0.92) for target lesions, nontarget lesions, and nonpathological lesions, respectively	The deep-learning-based semiautomated algorithm showed high accuracy for the treatment response assessment of pelvic lymph nodes and demonstrated comparable performance with radiologists.
Freely available artificial intelligence for pelvic lymph node metastases in PSMA PET-CT that performs on par with nuclear medicine physicians	Trägårdh E et al. [30]	2022	Eur J Nucl Med Mol Imaging	Retrospective	211	[18F]PSMA-1007 PET-CT	develop and validate a CNN for detection of PLNM in [18F]PSMA-1007 PET-CT from patients with high-risk PCa.	manual by 3 readers	Suspected pelvic lymph node metastases were marked by three independent readers vs. CNN	9	Sensitivity of the AI method for detecting pelvic lymph node metastases was 82%, and the corresponding sensitivity for the human readers was 77% on average.	This study shows that AI can obtain a sensitivity on par with that of physicians with a reasonable number of false positives.
Development and validation of the 3D U-Net algorithm for segmentation of pelvic lymph nodes on diffusion-weighted images	Liu X. et al. [21]	2021	BMC Med Imaging	Retrospective	393	3T mpMRI	evaluate the feasibility of the 3D U-Net algorithm for automated detection and segmentation of LNs on DWI	manual	Segmentation performance was assessed using the Dice score, positive predictive value (PPV), true positive rate (TPR), volumetric similarity (VS), Hausdorff distance (HD), the Average distance (AVD), and the Mahalanobis distance (MHD) with manual annotation of pelvic LNs as the reference. The accuracy with which the suspicious metastatic LNs (short diameter > 0.8 cm) were detected was evaluated using the area under the curve (AUC) at the patient level, and the precision, recall, and F1-score were determined at the lesion level.	11	The precision, recall, and F1-score for the detection of suspicious LNs were 0.97, 0.98, and 0.97, respectively. In the temporal validation dataset, the AUC of the model for identifying PCa patients with suspicious LNs was 0.963 (95% CI: 0.892–0.993).	The 3D U-Net algorithm can accurately detect and segment pelvic LNs based on DWI images.
Artificial intelligence-based detection of lymph node metastases by PET/CT predicts prostate cancer-specific survival.	Borrelli P. et al. [31]	2021	Clin Physiol Funct Imaging	Retrospective	399	PET/CT	developing and validating an AI-based tool for detection of lymph node lesions on PET/CT	automatically by Organ CNN	The tool consisted of CNN using complete PET/CT scans as inputs. In the test set, the AI-based LN detections were compared to those of two independent readers. The association with PCa-specific survival was investigated.	7	The AI-based tool detected more lymph node lesions than Reader B (98 vs. 87/117; *p* = 0.045) using Reader A as reference. AI-based tool and Reader A showed similar performance (90 vs. 87/111; *p* = 0.63) using Reader B as reference.	This study shows the feasibility of using an AI-based tool for automated and objective interpretation of PET/CT images that can provide assessments of lymph node lesions comparable with that of experienced readers, and prognostic information in PCa patients.

**Table 2 jcm-12-07032-t002:** Machine Learning algorithms and methods employed by each paper.

Title	Authors	Journal	Machine Learning Algorithms and Methods
Preoperative prediction of pelvic lymph nodes metastasis in prostate cancer using an ADC-based radiomics model: comparison with clinical nomograms and PI-RADS assessment.	Liu X. et al. [17]	Abdom Radiol (NY)	- U-Net: Used for automated segmentation of the prostate and prostate cancer lesions. - Radiomics Models: Four radiomics models were constructed based on different masks derived from the ADC map.
Utility of diffusion-weighted imaging-based radiomics nomogram to predict pelvic lymph node metastasis in prostate cancer.	Liu X. et al. [18]	BMC Med Imaging	- U-Net: for LN Segmentations, a deep learning algorithm was used to automatically identify and segment pelvic lymph nodes in diffusion-weighted imaging (DWI) images. - Logistic Regression Model: for Models 1 and 2, for PLNM prediction without and with clinical features. - Nomogram and Decision Curve Analysis (DCA): A nomogram based on Model 1 and Model 2 was developed to provide a visual prediction of the probability of PLNM.
Multiparametric MRI-based radiomics model to predict pelvic lymph node invasion for patients with prostate cancer	Zheng H. et al. [19]	Eur Radiol.	- Support Vector Machine (SVM) with a quadratic kernel was used to build the Integrative Radiomics Model (IRM) prediction model.
Development of a Radiomic-Based Model Predicting Lymph Node Involvement in Prostate Cancer Patients	Bourbonne V. et al. [20]	Cancers (Basel)	- Multilayer Perceptron Network: neural network used to develop a prediction model, combining clinical and radiomic features
Radiomics-based prognosis classification for high-risk prostate cancer treated with radiotherapy	Franzese C. et al. [25]	Strahlenther Onkol	A homemade radiomics model (type not specified) based on features extracted using the software package LifeX
A CT-based radiomics model to detect prostate cancer lymph node metastases in PSMA radioguided surgery patients	Peeken JC. et al. [26]	Eur J Nucl Med Mol Imaging	Two different modeling approaches: - Least Absolute Shrinkage Selection Operator (LASSO) - Logistic Regression
[18F]FMCH PET/CT biomarkers and similarity analysis to refine the definition of oligometastatic prostate cancer	Sollini M. et al. [32]	EJNMMI Res.	Did not provide explicit details about specific machine learning algorithms used for predictive modeling. It primarily discusses radiomics feature extraction, univariate analysis, similarity analysis, and unsupervised clustering.
Radiomic features from PSMA PET for non-invasive intraprostatic tumor discrimination and characterization in patients with intermediate- and high-risk prostate cancer–a comparison study with histology reference	Zamboglou C. et al. [27]	Theranostic	Extraction of radiomic features from PET/CT images. The machine learning algorithms used are not specified
Machine learning-based analysis of [18F]DCFPyL PET radiomics for risk stratification in primary prostate cancer	Cysouw MCF et al. [28]	Eur J Nucl Med Mol Imaging	- Random Forest
A machine-learning-assisted decision-support model to better identify patients with prostate cancer requiring an extended pelvic lymph node dissection	Hou Y. et al. [23]	BJU Int.	- Logistic Regression (LR) - Support Vector Machine (SVM) - Random Forests (RFs)
Integration of clinicopathologic identification and deep transferrable image feature representation improves predictions of lymph node metastasis in prostate cancer	Hou Y et al. [24]	EBioMedicine	- Random Forest Classifiers - Mean Decrease Gini Index (MDGI) - Sigmoid Function for probability conversion
Freely Available, Fully Automated AI-Based Analysis of Primary Tumour and Metastases of Prostate Cancer in Whole-Body [18F]-PSMA-1007 PET-CT.	Trägårdh E et al. [29]	Diagnostics (Basel)	- 3D Convolutional Neural Network (CNN) for the segmentation and classification of the context of [18F]PSMA PET-CT scans
Semiautomated pelvic lymph node treatment response evaluation for patients with advanced prostate cancer: based on MET-RADS-P guidelines	Liu X et al. [20]	Cancer Imaging.	- 3D U-Net segmentation used for segmentation
Freely available artificial intelligence for pelvic lymph node metastases in PSMA PET-CT that performs on par with nuclear medicine physicians	Trägårdh E et al. [30]	Eur J Nucl Med Mol Imaging	- 3D U-Net Convolutional Neural Network (CNN)
Development and validation of the 3D U-Net algorithm for segmentation of pelvic lymph nodes on diffusion-weighted images	Liu X. et al. [21]	BMC Med Imaging	- 3D U-Net Convolutional Neural Network (CNN)
Artificial-intelligence-based detection of lymph node metastases by PET/CT predicts prostate cancer-specific survival.	Borrelli P. et al. [31]	Clin Physiol Funct Imaging	- Organ CNN (convolutional neural network) - Detection CNN (convolutional neural network)

## 4. Discussion

In recent years, despite the extensive body of research on AI tools for the diagnosis of PCa, only a few studies have explored the use of radiomics models applied to imaging techniques for the detection and prediction of LNI in these patients. Specifically, the articles identified by our researchers were all published after 2019, predominantly between 2021 and 2022. Among these, only three studies are prospective, and five of these studies have enrolled fewer than 100 patients.

Despite the limited number of studies and small sample sizes, the preliminary findings from studies utilizing MRI are encouraging. Liu et al. utilized radiomics models based on ADC maps for predicting LNI in PCa, and their findings suggest that the proposed radiomics model performs at a comparable level of accuracy to the current standard models, when compared to commonly used nomograms such as Briganti 2017 and MSKCC nomograms and PI-RADS assessment models. Moreover, the research group also investigated the application of DWI-based radiomics, discovering that a combined prediction model, incorporating both radiological and radiomics features, enhances the accuracy of predicting LNI in PCa [17,18,20,21].

Furthermore, additional studies employing multiparametric MRI radiomics models have exhibited promising results in various aspects of prostate cancer (PCa) management. Zheng et al. proposed a radiomics model with an impressive AUC of 0.915 for predicting pelvic LNI in PCa patients. Bourbonne et al. developed a radiomics-based risk prediction model for LNI that outperformed existing models. Liu et al. achieved high accuracy (AUC of 0.963) in detecting suspicious lymph nodes using a 3D U-Net algorithm. Hou et al. utilized machine learning to improve the identification of patients requiring extended pelvic lymph node dissection and developed a PLNM-risk calculator with strong diagnostic accuracy [23]. These studies collectively demonstrate the potential of radiomics and deep learning approaches in enhancing LNI prediction, treatment response assessment, and risk prediction in PCa.

Our researchers identified two articles that employ CT radiomics models (one utilizing contrast-enhanced CT and the other utilizing non-contrast CT). In their study, Franzese et al. focused on evaluating the effectiveness of radiomics models derived from non-contrast CT treatment planning series in predicting metastatic progression in high-risk prostate cancer (PCa) patients [25]. The study compared the performance of pure clinical models, pure radiomics models, and a combined predictive model, the latter of which incorporated both clinical and radiomics features and exhibited superior predictive performance compared to the pure clinical and pure radiomics models. Contrast CT was used by Peeken JC et al. to develop a radiomics model to predict the presence of LNI in patients who underwent PSMA radioguided surgery to remove PSMA PET/CT-positive LNI; the proposed model showed a significantly better predictive performance compared to conventional CT parameters [26].

The utilization of AI tools applied to PET-CT images has also been recently studied, with encouraging findings. For example, it was found that radiomics features derived from PSMA PET were effective in distinguishing between cancerous and non-cancerous tissue in the prostate (Zamboglou et al.) [27]. Cysouw et al. used radiomics-based models using [18F]DCFPyL PET-CT to successfully predict lymph node involvement, metastasis, Gleason score, and extracapsular extension in intermediate- to high-risk PCa patients [28]. Trägårdh et al. developed an AI-based method (using [18F]PSMA-1007 PET-CT images) for detecting and quantifying PCa, lymph node metastases, and bone metastases, achieving higher average sensitivity compared to nuclear medicine physicians [29, 30]. Borrelli et al. developed an AI-based tool for detecting LN lesions in PCa patients using 18F-choline PET/CT scans, demonstrating similar performance to experienced readers and showing associations with PCa-specific survival [31].

In regard to the aims of the 16 works included, some evaluate the accuracy of the AI models in the identification and segmentation of lymph nodes, while the majority of them test their ability in pathological prediction. Again, two works evaluate the prognosis, and one, the assessment of the therapeutic response. What is interesting to note is that there is extreme variability among the works included, not only in terms of the imaging technique/sequence used and the objective but also and above all, in terms of the methodology applied and the technical aspects. In particular, most of the papers use traditional radiomics models while only five use a deep approach, of which two were applied to PET-CT images and three to MR images.

Segmentation in most cases is a manual segmentation, indicative of how there is still a need for standardization and automation of segmentation.

However, a notable practice of validation of the included set is evident, mainly through a testing group, in order to improve the quality of the results obtained, and considering that the samples analyzed are always relatively small (<500 patients).

The methodology employed varies considerably from work to work, starting from the type of features included (for example, only a few works, such as that of Liu X [21] compare the use of radiomics features vs. non-radiomics features), to the structure of the computational model (the only algorithms that reappear in different papers are Random Forest and Support Vector Machine). Consequently, it is evident that with these assumptions, a generalization of the results or a standardization of the methodology to be employed is not yet possible. However, the results obtained, with accuracies always higher than 85%, are extremely encouraging and suggest continued research in this area, especially justified by the high potential of these algorithms, which would allow increasingly less invasive approaches with ever-increasing reliability.

In particular, a significant variability in terms of objectives has been observed. Some studies assess the effectiveness of segmentation (automatic vs. manual vs. clinical nomograms) while others explore the analysis provided by the proposed model of the lesion or the entire gland. Differences among the studies also included the technique used (MRI vs. CT vs. PET/CT) and machine learning analysis. However, important insights can be summarized based on the results obtained. During the preparation of the dataset, it is crucial to establish a clear distinction between the training and test sets, ideally allocating about 60–80% to the training set. The most relevant selected features include texture, first-order statistical, and shape-based features, although wavelets have also yielded excellent results, as demonstrated by Franzese et al. Local Binary Pattern (LBP) features have been innovatively tested in the field of radiomics and were successfully utilized by Peeken et al., achieving a promising AUC of 95% [25,26]. Feature selection is important to mitigate overfitting issues. No discrepancies among readers were reported when assessing inter-reader variability, although the radiologist’s experience can slightly enhance model performance. To date, it is not possible to recommend the use of one radiomic model over another, as there is insufficient research to ensure the generalizability of results. However, the best performance has been achieved with logistic regression and support vector machine models. Very encouraging results have been also obtained with deep learning neural models, although the studies collected in this review differ in terms of input (lymph node or prostatic lesion) and imaging technique used (CT-PET vs. MRI). In this context, machine learning and deep neural networks require validation on external datasets to demonstrate the non-randomness of the results, particularly given the limited sample size involved.

Regarding the limitations of these studies, it is important to note the following:-Limited number of articles and retrospective design: the low number of articles analyzing LNI in PCa, coupled with the majority being retrospective in design, poses limitations in terms of sample size and potential biases in data collection.-Variability and lack of standardization: There is a lack of standardized approaches in various aspects, including the software used for analysis, as well as the type and number of features analyzed. This variability hinders the comparability and reproducibility of results. However, this extreme variability also indicates how many aspects are susceptible to evaluation and how much room there is for growth.-Manual segmentation and reproducibility: the reliance on manual segmentation for feature extraction introduces subjectivity and variability, leading to reduced reproducibility of results. The lack of standardized segmentation methods hampers the comparability of findings across different studies.

It is crucial to address these limitations and establish standardized methodologies to ensure reproducibility, comparability, and increased clinical relevance in future studies.

## 5. Conclusions

Overall, the findings of these studies suggest that AI-based models have the potential to improve the accuracy of detecting and predicting LN metastasis in PCa. These models can integrate clinical, radiological, and radiomics data to provide a more comprehensive assessment of LN involvement. The use of AI-based models can reduce unnecessary ePLND, spare patients from unnecessary morbidity, and improve patient outcomes. However, further studies are required to validate these models and determine their clinical utility in routine practice.

## Figures and Tables

**Figure 1 jcm-12-07032-f001:**
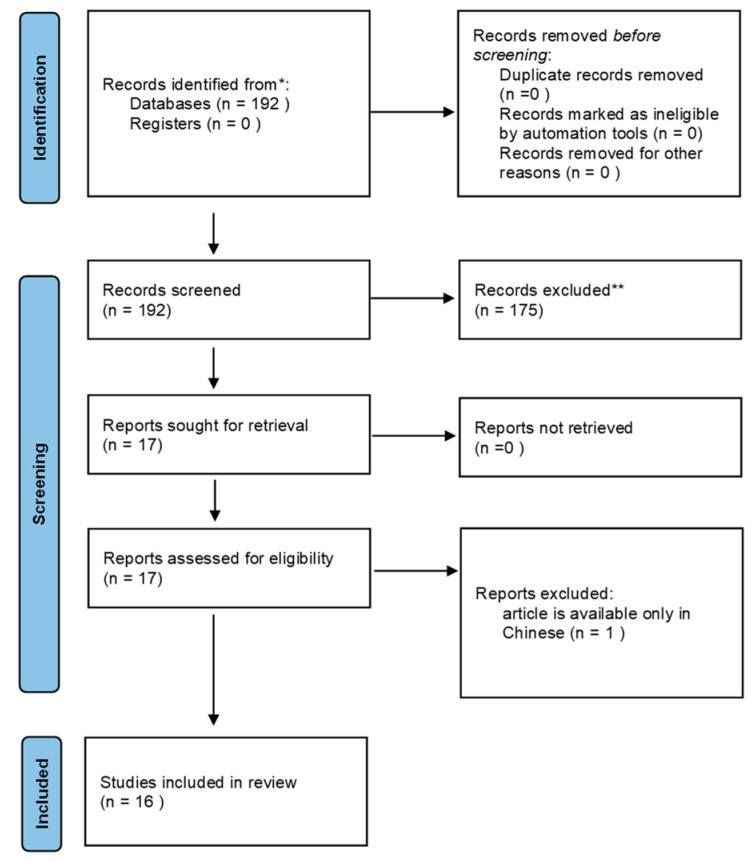
PRISMA 2020 flow diagram for the selection of studies included in the review. Inclusion (*) and exlusion (**) criteria have been reported in Methodology section.

## Data Availability

All data are available online.

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
