# Peer review of "Can Machine Learning Models Detect and Predict Lymph Node Involvement in Prostate Cancer? A Comprehensive Systematic Review"

_jcm, 2023, doi:10.3390/jcm12227032_

Round 1
Reviewer 1 Report
Comments and Suggestions for Authors
Faiella et al. present an interesting systematic review concerning the employment of imaging-guided AI and ML in predicting lymph node metastasis' presence (or absence) in men with prostate cancer diagnosis.
Here's my point-by-point analysis:
- Introduction: the introduction is well presented and structured. However, there are several typos in the text.
- Material and methods: this section needs to be extensively revised:
1) The authors must state they followed the PRISMA statement while writing the systematic review (or, if they did not, explain why).
2) Two authors should consult at least two databases independently.
3) Inclusion and exclusion criteria should be better explained.
4) did the authors follow the PICO framework when performing the research?
5) was the systematic review pre-registered on PROSPERO?
6) The search algorithm is not correct. A good alternative should be (prostate) AND (radiomics OR Machine OR Deep OR Artificial) AND (lymph).
7) Were the references of the included studies screened by searching for other eligible articles?
8) Which variables were extracted from the studies?
9) How quality assessment was performed?
10) The number of included studies should be reported in the results section.
Results: as for the material and methods section, the results section needs to be improved:
1) As already stated, the literature search results should be reported in the results section.
2) I miss a figure reporting the comprehensive overview of the study selection process (with the number of included and excluded articles).
3) General study characteristics should be better explained, and differences in index tests should be emphasized.
4) I miss the "Risk of Bias and Applicability" assessment.
Discussion: the discussion is well-written and structured. However, it just reports the results of single studies; I encourage the authors to revise it, discussing the most recent hypotheses and warranting analyses in a specific direction based on what they observed in their systematic review.
Finally, AI development is not based only on imaging, so I suggest the authors compare their observations with the performances of clinical-based algorithms used to predict the presence of lymph node metastases in prostate cancer patients (PMID: 37488275).
Comments on the Quality of English Language
The English is quite good. I suggest the authors to revise the text since I found several typos.
Author Response
REVIEWER 1 REPLY
Faiella et al. present an interesting systematic review concerning the employment of imaging-guided AI and ML in predicting lymph node metastasis' presence (or absence) in men with prostate cancer diagnosis.
Here's my point-by-point analysis:
- Introduction: the introduction is well presented and structured. However, there are several typos in the text.
- Thanks to the Reviewer for the comment. We modified the text and corrected the typos (we made the required changes in the entire manuscript without trying to trace them with Word's "revision" method. We apologize for the inconvenience)
- Material and methods: this section needs to be extensively revised:
1) The authors must state they followed the PRISMA statement while writing the systematic review (or, if they did not, explain why).
1) In the process of composing the review, we adhered to the PRISMA statement guidelines. The checklist is available for reference attached to this email, as supplementary materials, and the flowchart has been incorporated into the text.
2) Two authors should consult at least two databases independently.
2) Thanks to the Reviewer for this suggestion. We have increased the number of radiologists who independently consulted the databases to two.
3) Inclusion and exclusion criteria should be better explained.
3) We have improved and further clarified the inclusion and exclusion criteria in the text.
4) did the authors follow the PICO framework when performing the research?
4) No, the authors did not follow the PICO framework when performing the research.
5) was the systematic review pre-registered on PROSPERO?
- PROSPERO does not accept scoping review, literature reviews or mapping reviews, as reported in the following link: https://www.crd.york.ac.uk/PROSPERO/#registernew.
6) The search algorithm is not correct. A good alternative should be (prostate) AND (radiomics OR Machine OR Deep OR Artificial) AND (lymph).
6) Thanks for your suggestion. We have made the recommended changes to the text.
7) Were the references of the included studies screened by searching for other eligible articles?
7) Yes, but no other articles that fully met the inclusion criteria of this review were found.
8) Which variables were extracted from the studies?
8) The extracted variables are reported in the table 1. We highlighted them adding the table caption in the text.
9) How quality assessment was performed?
9) Quality assessment was conducted using the Radiomics Quality Score (RQS) to evaluate the methodological quality (which can be found in the supplementary table) as reported in material method section. We stressed this concept in the M&M section.
10) The number of included studies should be reported in the results section.
10) The number of included studies was reported in results section.
Results: as for the material and methods section, the results section needs to be improved:
1) As already stated, the literature search results should be reported in the results section.
1) The results of included studies have been reported in results section.
2) I miss a figure reporting the comprehensive overview of the study selection process (with the number of included and excluded articles).
2) We have added the PRISMA flowchart to the text, which describes the study selection process.
3) General study characteristics should be better explained, and differences in index tests should be emphasized.
3) We have made some modifications to the text for this purpose.
4) I miss the "Risk of Bias and Applicability" assessment.
4) Risk of Bias was assessed as part of the calculation of the Radiomics Quality Score for each included study
Discussion: the discussion is well-written and structured. However, it just reports the results of single studies; I encourage the authors to revise it, discussing the most recent hypotheses and warranting analyses in a specific direction based on what they observed in their systematic review.
Finally, AI development is not based only on imaging, so I suggest the authors compare their observations with the performances of clinical-based algorithms used to predict the presence of lymph node metastases in prostate cancer patients (PMID: 37488275).
Thank you for the suggestions. We have made an effort to enhance the text as recommended.
In the discussion, clinical aspects and clinical-based algorithms are also mentioned. Despite the limited number of studies and small sample sizes, the preliminary findings from studies utilizing MRI are encouraging. For instance, Liu et al. utilized radiomics models based on ADC maps to predict LNI in PCa. Their findings suggest that the proposed radiomics model performs at a comparable level of accuracy to current standard models, as evidenced by comparisons with commonly used nomograms such as Briganti 2017, MSKCC nomograms, and PI-RADS assessment models.
Comments on the Quality of English Language
The English is quite good. I suggest the authors to revise the text since I found several typos.
We corrected the typos and English errors in the text. (we made the required changes in the entire manuscript without trying to trace them with Word's "revision" method. We apologize for the inconvenience)

Reviewer 2 Report
Comments and Suggestions for Authors
This article intriguingly focuses on comparing the applications of machine learning models for prostate cancer detection/prediction based on lymph node involvement, taking into account some of the latest publications. This is an interesting and important research field. However, an area that requires enhancement is the detailed discussion of the specific machine learning models employed in each referenced study. Although the results are presented in this review, there's scant information concerning the distinct models used in each previous publication. It would be especially enlightening to ascertain if the outcomes reported in each published work were a direct result of the machine learning model in question. A thorough comparison of the machine learning models, as well as the datasets used for training and testing, would be valuable. Such insights would assist readers in refining their future designs of machine learning models and the selection of datasets for training and testing.
Author Response
REVIEWER 2 REPLY
This article intriguingly focuses on comparing the applications of machine learning models for prostate cancer detection/prediction based on lymph node involvement, taking into account some of the latest publications. This is an interesting and important research field. However, an area that requires enhancement is the detailed discussion of the specific machine learning models employed in each referenced study. Although the results are presented in this review, there's scant information concerning the distinct models used in each previous publication. It would be especially enlightening to ascertain if the outcomes reported in each published work were a direct result of the machine learning model in question. A thorough comparison of the machine learning models, as well as the datasets used for training and testing, would be valuable. Such insights would assist readers in refining their future designs of machine learning models and the selection of datasets for training and testing.
Thanks to the Reviewer for this suggetsion which allow us to improve our manuscript and make it more intersting also for a technical audience.
We added the following paragraph:
Even more, a significant variability in terms of objectives has been observed. Some studies assess the effectiveness of segmentation (automatic vs. manual vs. clinical nomograms) while others explore the analysis provided by the proposed model of the lesion or the entire gland. Differences among the studies included also the technique used (MRI vs. CT vs. PET/CT) and machine learning analysis. However, important insights can be summarized based on the results obtained. During the preparation of the dataset, it is crucial to establish a clear distinction between the training and test sets, ideally allocating about 60-80% to the training set. The most relevant selected features include texture, first-order statistical, and shape-based features, although wavelets have also yielded excellent results, as demonstrated by Franzese et al. Local Binary Patterns (LBP) features have been innovatively tested in the field of radiomics and were successfully utilized by Peeken et al., achieving a promising AUC of 95%26. Features selection is important to mitigate overfitting issues. No discrepancies among readers were reported when assessing inter-reader variability, although the radiologist's experience can slightly enhance model performance. To date, it is not possible to recommend the use of one radiomic model over another, as there is insufficient research to ensure the generalizability of results. However, the best performance has been achieved with logistic regression and support vector machine models. Very encouraging results have been also obtained with deep learning neural models, although the studies collected in this review differ in terms of input (lymph node or prostatic lesion) and imaging technique used (CT-PET vs. MRI). In this context, machine learning and deep neural networks require validation on external datasets to demonstrate the non-randomness of the results, particularly given the limited sample size involved.

Round 2
Reviewer 1 Report
Comments and Suggestions for Authors
The authors accomplished an excellent manuscript revision, answering reviewers' requests.
Author Response
Thanks to the Reviewer